# Association of Klotho with Coronary Artery Disease in Subjects with Type 2 Diabetes Mellitus and Preserved Kidney Function: A Case-Control Study

**DOI:** 10.3390/ijms241713456

**Published:** 2023-08-30

**Authors:** Javier Donate-Correa, Ernesto Martín-Núñez, Carmen Mora-Fernández, Ainhoa González-Luis, Alberto Martín-Olivera, Juan F. Navarro-González

**Affiliations:** 1Unidad de Investigación, Hospital Universitario Nuestra Señora de Candelaria, 38010 Santa Cruz de Tenerife, Spain; emarnu87@gmail.com (E.M.-N.); carmenmora.fdez@gmail.com (C.M.-F.); ainhoa.gonaluz@gmail.com (A.G.-L.); olimarabe@gmail.com (A.M.-O.); 2GEENDIAB (Grupo Español para el Estudio de la Nefropatía Diabética), Sociedad Española de Nefrología, 39008 Santander, Spain; 3Instituto de Tecnologías Biomédicas, Universidad de La Laguna, 382500 Santa Cruz de Tenerife, Spain; 4RICORS2040 (RD21/0005/0013), Instituto de Salud Carlos III, 28029 Madrid, Spain; 5Servicio de Nefrología, Hospital Universitario Nuestra Señora de Candelaria, 38010 Santa Cruz de Tenerife, Spain

**Keywords:** type 2 diabetes mellitus, coronary artery disease, Klotho

## Abstract

Circulating Klotho levels are significantly reduced in subjects with type 2 diabetes mellitus (T2DM) and in kidney disease patients. In this work, the relationship between Klotho levels and the coronary artery disease (CAD) burden in subjects with T2DM and preserved kidney function was analyzed. For this, we performed a cross-sectional case-control study involving 133 subjects with T2DM and 200 age-, sex- and CAD-incidence-matched, non-diabetic patients undergoing non-emergency diagnostic coronary angiography. All of them were non-albuminuric and with normal glomerular filtration rates. The concentrations of serum Klotho, fibroblast growth factor 23, and inflammatory markers were also measured. As expected, the serum Klotho concentration was lower in the T2DM group (12.3% lower, *p* = 0.04). However, within the group of patients with T2DM, those subjects with CAD presented significantly higher Klotho levels than those without significant coronary stenosis (314.5 (6.15–562.81) vs. 458.97 (275.2–667.2) pg/mL; *p* = 0.02). Multiple regression analysis revealed that serum Klotho was positively related with stenosis values exclusively in subjects with T2DM (adjusted R2 = 0.153, *p* < 0.01). Moreover, logistic regression analysis showed that Klotho was positively associated with the presence of significant CAD in the group of T2DM patients (OR: 1.001; *p* = 0.041). Our data suggest that higher levels of circulating Klotho in subjects with T2DM and preserved kidney function are associated with the presence of significant CAD.

## 1. Introduction

Type 2 diabetes mellitus (T2DM) accounts for over 90 per cent of all subjects with diabetes [1]. These patients present multiple comorbidities, especially those with poorly controlled blood glucose levels, high blood pressure, and dyslipidemia, vascular complications being the main cause of morbidity and mortality [2]. Coronary artery disease (CAD), a main macrovascular diabetic complication and a leading cause of morbidity and mortality, is mediated primarily by atherosclerosis, a multifactorial disorder in which chronic inflammation, together with oxidative stress and endothelial dysfunction, are key underlying processes [3,4]. CAD is often asymptomatic in T2DM subjects until severe disease develops with the presentation of myocardial infarction or sudden cardiac death [5]. Therefore, novel biomarkers for detecting CAD at subclinical stages are increasingly necessary in order to enable the earliest possible therapeutic interventions and to characterize new therapeutic targets.

Klotho is a transmembrane protein highly expressed in the kidneys where it functions as a coreceptor for the signal transduction of fibroblast growth factor 23 (FGF23), a bone-derived hormone that regulates urinary phosphate excretion [6]. The extracellular domain of membrane Klotho can be shed by secretases and released into the circulation as a soluble form [7]. Soluble Klotho is able to weakly transduce the FGF23 signaling pathway but also functions as an enzyme and hormonal factor to play a variety of biological functions. The decreased expression of Klotho is closely associated with chronic kidney disease (CKD) [8], but reduced Klotho levels are also observed in T2DM subjects with preserved kidney function [9,10]. The previous studies have associated these reductions in Klotho with the increased levels of inflammatory markers, as well as with a higher incidence of vascular disease and subclinical atherosclerosis both in subjects with CKD and in the general population [11,12,13,14]. This observation has led us to consider the reduction in Klotho expression as a predictor of long-term macrovascular outcomes in T2DM [15]. However, clinical studies examining the possible association of Klotho with vascular complications in diabetes have reported inconsistent results [16,17,18,19]. Moreover, the relationship, if any, of Klotho with subclinical CAD burden and inflammation in subjects with T2DM in the absence of kidney impairment, has not been assessed yet.

In view of this, we conducted this observational study to investigate the associations of circulating levels of Klotho with CAD and with inflammatory parameters in T2DM subjects who were free of kidney impairment and previous cardiovascular events.

## 2. Results

The demographic, clinical and biochemical data of subjects with and without T2DM are shown in Table 1. Four hundred and ninety-one patients were considered for enrolment in the study, and 158 were excluded due to the exclusion criteria. Therefore, 333 patients (220 male) were finally included (Table 1). One hundred and thirty-three subjects had T2DM. The mean age was 65.3 ± 10.9 years and the median BMI was 26.1 ± 3.2 kg/m^2^. There was no difference in demographics and comorbidities between the two groups except for the prevalence of current smokers, which was higher in the group of non-T2DM subjects (31.5% vs. 23.2%, *p* = 0.043). Significant CAD was present in 270 patients, with a similar prevalence in non-T2DM and T2DM subjects (83.5% vs. 77.4%, *p* = 0.17, respectively). Serum total cholesterol concentrations were also similar in both groups. Conversely, triglycerides (TG), triglyceride-glucose index (TyG), HDL- and LDL-cholesterol levels were lower in the subjects with T2DM; these differences might be explained by the higher percentage of subjects with T2DM treated with lipid lowering medications (52% vs. 63%, *p* = 0.04). The mean eGFR was 99.5 ± 3.3 mL/min/1.73 m^2^, without differences between groups. ACR was higher in subjects with T2DM, although this difference did not reach statistical significance (5.82 [3.54–15.9] vs. 8.38 [4.08–16.96] mg/g; *p* = 0.08). None of the different inflammatory parameters analyzed, including inflammatory cytokines and the NLR or MLR, differed between groups. Of note, the levels of serum Klotho were significantly reduced (480.8 [332.2–651] vs. 421.9 [252–651.4] pg/mL; *p* = 0.04) in subjects with T2DM (Table 1 and Figure 1A), whereas iFGF23 concentration did not differ between groups. When considering the differences in the stenosis of the epicardial arteries evaluated, the mean percentage of obstruction was similar in subjects with and without T2DM, except for the RCA, which was significantly higher in the group with T2DM (Table 2).

The sub-analysis of the group of subjects with T2DM attending to the occurrence of CAD revealed that Klotho was significantly higher in subjects with significant CAD (Klotho percent increase: 45.6%; *p* = 0.02) (Table 3 and Figure 1B), without differences in non-T2DM subjects (Figure 1C). The subjects with T2DM and CAD also presented higher SBP values (*p* = 0.04), higher concentrations of serum inflammatory parameters (hs-CRP, TNFα, and IL6), and a trend to increased levels of ACR. No differences were observed in the TyG index between T2DM subjects according to the presence of significant CAD.

A bivariate correlation analysis showed that Klotho was significantly and positively related with SSI values in the sub-group of T2DM patients (r = 0.223, *p* < 0.01), particularly with the percentage of stenosis observed in the two major epicardial arteries: LCA (r = 0.212, *p* = 0.01) and RCA (r = 0.182, *p* = 0.04) (Table 4). Klotho was also inversely related with MLR values (r = −0.184, *p* = 0.03). Importantly, these correlations were absent in the group of non-T2DM subjects.

The correlation between Klotho and SSI was also observed even after additional adjustment for potential confounders: age, sex, HT, current smokers, dyslipidemia, ACR, body mass index (BMI), phosphorus, iFGF23, TNFα, IL6, and hs-CRP. Thus, the results of a forward stepwise multiple regression analysis performed with the SSI as the dependent variable showed the independent and positive association between the severity of stenosis with soluble Klotho, dyslipidemia and ACR (adjusted R2 = 0.153, *p* < 0.01) (Table 5). Again, the association between SSI and Klotho was not present in the group of non-T2DM subjects, where TNFα and ACR were the variables significantly and independently associated with the SSI (adjusted R2 = 0.045, *p* < 0.05). Notably, the ACR remained significantly associated with SSI in both groups.

Finally, we performed a multivariate logistic regression analysis for the presence of significant CAD as the dependent variable and serum Klotho levels as an independent variable (Table 6). Traditional risk factors for CAD (age, HT, current smokers, and dyslipidemia) were entered as covariates (model 1). The model was additionally adjusted for markers of kidney function (eGFR and ACR) and iFGF23 (model 2) and inflammatory variables (MLR, hs-CRP, IL6 and TNFα) (model 3). The results showed that Klotho was positively associated with CAD, even after additional adjustment for all the potential confounders (OR [95% CI]: 1.001 [1.0–1.003], *p* < 0.05) (Table 6).

## 3. Discussion

The results of this study show reduced levels of serum Klotho in T2DM patients when compared to a group of non-T2DM subjects with similar age, sex distribution, BMI and comorbidities—including CAD—as well as preserved kidney function (eGFR ≥ 60 mL/min/1.73 m^2^) and normal albuminuria (ACR < 30mg/g). Counterintuitively, serum Klotho was positively associated with the occurrence and severity of CAD in the group of subjects with T2DM, independently of traditional risk factors for cardiovascular disease. This association was absent in the group on non-T2DM subjects. The group of patients with T2DM and significant CAD also presented higher levels of inflammatory parameters—including TNFα, IL6 and hsPCR—when compared with those without CAD.

Our observations contrast with the previous cross-sectional and longitudinal studies that showed an association between reduced serum Klotho levels and the presence and severity of CAD [13,15,20,21,22,23,24]. Klotho is predominantly expressed in the kidneys, and in humans, the secreted form of Klotho is more dominant than the membrane form [25]. Consistent with our results, serum concentration of this hormone was shown to be reduced in T2DM subjects [16,26]. In CKD, Klotho deficiency appears in the early stages and gradually decreases with the progression of the disease [25,27,28], also being related to the appearance of vascular disorders, including arterial stiffness [12,23,29]. The inverse correlation between soluble Klotho levels and the existence of vascular disease, generally accepted in patients with CKD [12], does not seem to be as evident in subjects with diabetes. Previous clinical studies on this relationship report inconsistent results [16,17,18,19,30]. In a longitudinal study by Pan et al. [15], the authors observed a significant association between lower concentrations of circulating Klotho and an increased risk of developing long-term atherosclerotic cardiovascular disease (CAD and stroke). Unlike this study, other cross-sectional investigations suggest that the inverse association of Klotho levels with CAD disappears or has the opposite direction in subjects with preserved kidney function [17,19,31]. Ark et al. [17] found no significant differences in Klotho levels according to the presence of macrovascular disease (CAD or peripheral artery disease) in a small group of subjects with T2DM, preserved eGFR and ACR < 300 mg/g. In a study by Koga et al. [31] including patients with stable CAD, serum Klotho concentration was inversely related with the values of coronary artery calcification index only in the subgroup of subjects with CKD, being absent in those with preserved renal functionality. Interestingly, other studies that enrolled patients with moderately impaired kidney function, mostly patients with diabetes (74%), reported positive correlations of serum Klotho with matrix calcium deposition in coronary biopsies [32]. Furthermore, in subjects with type 1 diabetes mellitus and preserved kidney function, the association of Klotho with vascular disease was reported to be absent for arterial calcification [33] and to be positive for subclinical carotid vascular disease [19].

The discrepancy in the results evidences a more complex role of Klotho in vascular disease. Therefore, the association of Klotho with vascular disorders in patients with diabetes appears to differ depending on the specific disease conditions, particularly renal health status. In T2DM subjects, the decrease in serum Klotho has been related with the progression of kidney disease even from the early stages. In these patients, ACR independently and negatively correlates with serum Klotho concentrations even with low levels of urinary albumin [16], indicating the existence of early renal damage even before a clinically significant increase in albuminuria occurs. In these early stages of kidney disease there is hardly any loss of tubular cells to justify the decrease in Klotho, which is probably due to the metabolic and hemodynamic stress that occurs during diabetic disease. These stressors cause cellular damage and dysfunction and stimulate the onset of an inflammatory response and the subsequent fibrosis that results in kidney injury. Klotho protects cells against accelerated aging and damage during the course of diabetes and diabetic nephropathy by several mechanisms, including the inhibition of inflammation [34,35,36]. In fact, Klotho was also found to be related to multi-system inflammation response, and its preservation was reported as a potential adaptive kidney protective response in proximal tubular cells that limits the inflammatory response and the severity of tubular cell injury during acute kidney injury (AKI) [37]. The anti-atherogenic properties of Klotho were also attributed to this anti-inflammatory effect [14,29]. In view of this, it is possible to hypothesize that the association of Klotho with CAD in our group of patients is a counterregulatory mechanism to provide vascular protection via anti-inflammatory effects that is only present when the kidney function is normal, and therefore, the capacity to produce Klotho in this organ is preserved. In our study, we specifically analyzed only T2DM subjects with normal kidney function (preserved eGFR and ACR < 30 mg/g) and without previous cardiovascular events. Moreover, our results show elevated levels of inflammatory markers, including hs-CRP, TNFα and IL6, in T2DM subjects with significant CAD. However, none of these parameters correlated with serum Klotho levels, nor in non-T2DM or in T2DM subjects. Only MLR values were inversely correlated with circulating Klotho in T2DM. Moreover, a logistic regression analysis did not find any significant association between inflammatory parameters and the presence of significant CAD.

We acknowledge several limitations. First, the observational and cross-sectional design of the present study precludes us from making any causal inference. Potential confounding factors related to the increment in Klotho in patients with significant CAD might be unconsidered, such as calcitriol and parathyroid hormone levels. In addition to kidney function, several extrinsic factors including medication can influence serum Klotho concentrations counterbalancing its potential reductions in vascular disease. Experimental studies have shown positive correlations between treatment with statins and angiotensin II receptor blockers with Klotho expression [38,39]. Although in our study, the proportion of subjects with T2DM under treatment with statins and blockers of the renin-angiotensin system was similar in the groups with significant and not-significant CAD, we cannot exclude the adherence to medications as a contributing factor to the increment in Klotho levels. Moreover, insulin promotes the cleavage of Klotho from the plasmatic membrane and the generation of the soluble form [40]. Therefore, the administration of insulin might also modify serum concentration levels of Klotho in diabetic patients. Furthermore, the lack of a standardized assay to measure soluble Klotho might lead to inconsistent or even contradictory results from study to study. Finally, we are aware that this study raises important questions about the clinical significance of the relationship between Klotho and the presence of significant CAD in subjects with T2DM. An interesting question, particularly in subjects with T2DM, that has not yet been studied in depth is the potential relationship between the gut microbiota and Klotho. The gut microbiota has emerged as a significant contributory factor to age-related disease [41]. Moreover, the dysregulation in the composition and function of gut microbiota in patients with symptomatic atherosclerosis has been reported in many studies [42]. This dysregulation is proposed to trigger the immune system and cause systemic and local inflammation [43]. According to our hypothesis, this could cause a compensatory increase in Klotho production in patients with preserved kidney function. In any case, our results require subsequent validation in a much bigger cohort. The controversy about the utility of Klotho as a suitable biomarker of subclinical vascular disease remains and represents an area of ongoing research.

## 4. Materials and Methods

### 4.1. Study Design and Population

This cross-sectional study was designed to evaluate the association of serum Klotho levels with the occurrence and severity of CAD in T2DM subjects. The selected participants included 133 T2DM, and 200 age-, sex- and CAD-incidence-matched, non-diabetic consecutive patients undergoing nonemergency diagnostic evaluation for CAD by elective coronary angiography. The inclusion criteria were age > 18 years old, T2DM duration of at least 1 year, no evidence of previous cardiovascular disease, normal kidney function (defined as an estimated glomerular filtration rate (eGFR) > 60 mL/min/1.73 m^2^) and normal urinary albumin excretion (defined as a urine albumin-to-creatinine ratio (ACR) < 30 mg/g). The exclusion criteria were previous myocardial infarction, coronary angioplasty, intracoronary stent placement or coronary artery bypass graft surgery (CABG), peripheral vascular disease, cardiac arrhythmia, immunologic or inflammatory diseases (such as rheumatoid arthritis, systemic lupus erythematosus, or inflammatory bowel disease). In addition, we also excluded non-diabetic subjects with fasting glucose and HbA1c values upper to 100 mg/dL and 5.7%, respectively. No patient was receiving calcium, phosphate or vitamin D supplementation. All the protocols complied with the ethical standards of the Declaration of Helsinki and were reviewed and approved by the institutional ethics committee. Written informed consent was obtained from all participants.

### 4.2. Coronary Angiography

Coronary stenosis was determined by angiography using standard techniques. The assessment of stenosis was determined in four major epicardial arteries: left main coronary artery (LCA), left anterior descending artery (LAD), circumflex artery (CA) and right coronary artery (RCA). Significant CAD was defined as the presence of at least one lesion leading to ≥50% luminal diameter narrowing in any of the considered arteries. Additionally, a stenosis severity index (SSI) was defined as the average of the maximum stenosis values determined in each of those arteries [13].

### 4.3. Clinical and Biochemical Variables

The demographic and clinical data were obtained from each participant. General biochemical analyses using standard laboratory methods were performed in fasting blood and urine samples before angiography. Serum Klotho was measured by the specific enzyme-linked immunosorbent assay (ELISA) human Klotho kit (Immuno-Biological Laboratories, Takasaki, Japan) with a sensitivity of 6.15 pg/mL and intra- and inter-assay coefficients of variation of 3.1% and 6.9%, respectively. The serum levels of intact FGF23 were also determined by ELISA (EMD Millipore Corporation, Milford, MA, USA), which has a sensitivity of 3.5 pg/mL and intra- and inter-assay coefficients of 9.5% and 6.85%, respectively. The levels of inflammatory cytokines tumor necrosis factor-alpha (TNFα) α and interleukin (IL) 6 were measured by high-sensitivity ELISA kits (Quantikine^®^, R&D Systems, Abingdon, UK). The minimum detectable concentrations were 0.10 pg/mL and 0.50 pg/mL, respectively. The intra- and inter-assay coefficients of variability were <10.8%. High-sensitivity serum C-reactive protein (hs-CRP) was measured by a high-sensitivity particle enhanced immunoturbidimetric fully automated assay (Roche Diagnostics GmbH, Mannheim, Germany) in a Cobas 6000 analyzer from the same manufacturer with a sensitivity of 0.3 mg/L and intra- and inter-assay coefficients of variation of 1.6% and 8.4%, respectively. Neutrophil- and monocyte-to-lymphocyte ratios (NLR and MLR) were calculated for each patient by dividing the absolute number of neutrophils or monocytes by the absolute number of lymphocytes. We also determined the triglyceride–glucose (TyG) index, calculated as TyG index = ln [Fasting triglyceride (mg/dL) × fasting glucose (mg/dL)]/2, as a potential marker of vascular dysfunction.

### 4.4. Statistical Analysis

The quantitative variables were presented as mean and standard deviation or median and interquartile intervals, and the categorical data were expressed as frequencies and percentages. The comparisons between groups were performed by Chi-square test, Student’s *t*-test or Mann–Whitney U test, as appropriate. The Spearman correlation coefficient was calculated to assess the relation between soluble Klotho concentrations and other variables, including those obtained in coronary angiographic studies. A forward stepwise multiple regression analysis was performed to determine the independent association between potential predictor variables and the severity of CAD expressed as SSI. A multiple logistic regression was also performed for the presence of significant CAD as the dependent and serum Klotho levels as an independent variable. For this purpose, we adopted three models including those variables that showed significant correlations with Klotho levels and those that constitute traditional or new risk factors for CAD: in model 1, we introduced age, hypertension, current smokers and dyslipidemia; in model 2, we additionally included MLR, hs-CPR, TNFα and IL6; finally, in model 3, we adjusted the analysis for serum iFGF23, eGFR and ACR. All analyses were performed using SPSS software version 25 (IBM Corp., Armonk, NY, USA). A 2-tailed *p*-value less than 0.05 was considered statistically significant.

## 5. Conclusions

Silent coronary disease is common in patients with T2DM. Our data suggest that higher levels of circulating Klotho in subjects with T2DM and preserved kidney function could be a marker of significant CAD. If confirmed, these results could point to the existence of a new mechanism involved in the development of this pathology in T2DM patients.

## Figures and Tables

**Figure 1 ijms-24-13456-f001:**
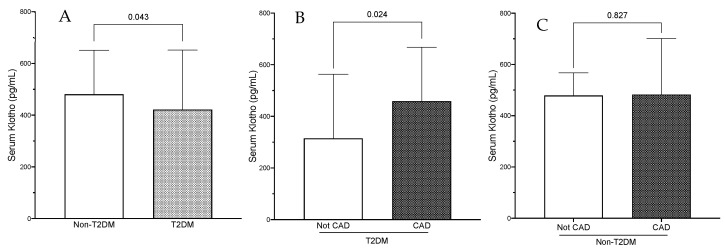
(**A**) Serum Klotho levels in the whole population according to T2DM status. (**B**,**C**) Serum Klotho levels according to the presence of CAD in patients with and without T2DM, respectively. The data are presented as the median and interquartile range (IQR). T2DM, type 2 diabetes mellitus; CAD, coronary artery disease.

**Table 1 ijms-24-13456-t001:** Demographic and biochemical characteristics of patients included in the study.

	Non-T2DM	T2DM	*p*	All Subjects
Characteristics				
N	200	133		333
Age (years)	64.6 ± 11.3	66.3 ± 10.2	0.27	65.3 ± 10.9
Sex (% male)	135 (67.5)	85 (63.9)	0.49	220 (66.1)
BMI (kg/m^2^)	25.3 ± 2.8	27.1 ± 2.1	0.09	26.1 ± 3.2
SBP (mm Hg)	123 ± 15.1	128 ± 11.5	0.12	127 ± 12.3
DBP (mm Hg)	74.5 ± 8.6	75.8 ± 9.7	0.13	74.9 ± 9.1
Comorbidities (%)				
Significant CAD	167 (83.5)	103 (77.4)	0.17	270 (81.1)
Obesity	28 (14)	14 (10.5)	0.35	42 (12.6)
Hypertension	61 (30.5)	37 (27.8)	0.59	98 (29.4)
Former smokers	40 (20)	28 (21.1)	0.46	68 (20.4)
Current smokers	63 (31.5)	21 (23.3)	0.04	94 (28.2)
Dyslipidemia	88 (44)	54 (40.6)	0.54	142 (42.6)
Laboratory data				
T-cholesterol (mg/dL)	172.6 (145–201.5)	168 (140–190)	0.14	171 (143–198)
HDL-C (mg/dL)	40.5 (33–48)	37 (31–43.5)	0.05	38 (32–46)
LDL-C (mg/dL)	103 (80–127)	94 (74.5–112)	<0.01	100.4 (76–121.8)
TG (mg/dL)	125 (92.3–162.8)	142 (114.5–197.5)	<0.001	135 (102–172)
TyG index	4.68 (4.53–4.79)	5.04 (4.9–5.25)	<0.001	4.8 (4.61–5.02)
FG (mg/dL)	90 (86–95)	158 (138–200)	<0.001	97 (89–147.5)
Hb1ac (%)	5.37 ± 0.37	7.57 ± 0.42	<0.001	6.28 ± 1.46
eGFR (mL/min/1.73 m^2^)	101.1 ± 9.2	97.5 ± 8.3	0.11	99.5 ± 3.3
Creatinine (mg/dL)	0.89 (0.75–1.05)	0.82 (0.7–1.02)	0.04	0.86 (0.74–1.04)
ACR (mg/g)	5.82 (3.54–15.9)	8.38 (4.08–16.96)	0.08	6.83 (3.68–16.04)
Uric acid (mg/dL)	5.54 (4.58–6.69)	5.36 (4.46–6.21)	0.29	5.43 (5.5–6.6)
Calcium (mg/dL)	9.32 ± 0.31	9.33 ± 0.38	0.56	9.32 ± 0.34
Phosphate (mg/dL)	3.54 ± 0.53	3.55 ± 0.67	0.71	3.54 ± 0.71
NLR (/mL)	2.3 (1.69–3.24)	1.78 (2.32–3.61)	0.28	2.31 (1.74–3.33)
MLR (/mL)	0.33 (0.26–0.49)	0.32 (0.24–0.44)	0.06	0.33 (0.24–0.46)
hs-CRP (mg/L)	3.2 (1.92–6.23)	3.3 (2–6.7)	0.74	3.2 (2–6.6)
TNFα (pg/mL)	1.99 (1.35–2.55)	2.09 (1.52–2.99)	0.12	2.06 (1.43–2.77)
IL6 (pg/mL)	6.12 (3.49–10.35)	6.36 (3.3–12.14)	0.52	6.3 (3.3–10.76)
Klotho (pg/mL)	480.8 (332.2–651)	421.9 (252–651.4)	0.04	466.3 (290.5–651.4)
iFGF23 (pg/mL)	583.3 (471.7–790.1)	570.7 (427.3–706.1)	0.11	577 (443.1–766.1)
Medication				
Statin (%)	104 (52)	84 (63.2)	0.04	188 (56.5)
ACEI/ARB (%)	80 (40)	61 (45.9)	0.29	141 (42.3)

T2DM, type 2 diabetes mellitus; BMI, body mass index; SBP, systolic blood pressure; DBP, diastolic blood pressure; CAD, coronary artery disease; HDL-C high density lipoprotein cholesterol; LDL-C low-density lipoprotein cholesterol; TG, triglycerides; TyG, triglyceride glucose index; FG, fasting glucose; Hb1ac, glycated hemoglobin; eGFR, estimated glomerular filtrate rate; ACR, urine albumin-to-creatinine ratio; hs-CRP high sensitivity C-reactive protein, TNFα, tumor necrosis factor alpha; IL, interleukin; NLR, neutrophil lymphocyte ratio; MLR, monocyte lymphocyte ratio; iFGF23, intact fibroblast growth factor 23; ACEI, angiotensin converting enzyme inhibitor; ARB, angiotensin II receptor antagonist.

**Table 2 ijms-24-13456-t002:** Coronary angiography study results.

	Non-T2DM	T2DM	*p*	All Subjects
SSI	30.13 (16.1–44.94)	33.5 (16.13–54.13)	0.39	32.25 (16.1–51.13)
Obstruction (%)				
LCA	55.12 ± 29.1	54.2 ± 29.21	0.61	54.75 ± 29.1
RCA	25.86 ± 34.66	33.16 ± 37.4	0.02	28.78 ± 35.91
LAD	33.43 ± 38.92	34.59 ± 40.05	0.33	33.89 ± 39.32
CA	14.54 ± 30.82	14.77 ± 31.1	0.94	14.63 ± 30.89

T2DM, type 2 diabetes mellitus; SSI, severity stenosis index; LCA, left main coronary artery; RCA, right coronary artery; LAD, left anterior descending artery; CA, circumflex artery.

**Table 3 ijms-24-13456-t003:** Sub-analysis of demographic and biochemical characteristics of subjects with T2DM according to the presence of significant CAD.

	No CAD	CAD	*p*
Characteristics			
N	30	103	
Age (years)	63 (54.8–73)	68 (61–75)	0.16
Sex (% male)	20 (66.7)	65 (63.1)	0.83
BMI (kg/m^2^)	26.8 ± 2.2	27.3 ± 2.9	0.13
SBP (mm Hg)	129 ± 10.1	132 ± 11.1	0.04
DBP (mm Hg)	75.3 ± 7.7	75.9 ± 9.1	0.47
Comorbidities			
Obesity	4 (13.3)	10 (9.7)	0.52
Hypertension (%)	6 (20)	31 (30.1)	0.36
Former smoker (%)	7 (23.3)	21 (20.4)	0.12
Current smokers (%)	7 (23.3)	19 (18.4)	0.21
Dyslipidemia (%)	10 (33)	44 (42.7)	0.41
Laboratory data			
T-cholesterol (mg/dL)	160.5 (134.5–188.5)	168 (142–193)	0.47
HDL-C (mg/dL)	34 (27.3–42.8)	38 (31–44)	0.26
LDL-C (mg/dL)	90.5 (70.3–118.5)	95 (75–110)	0.84
TG (mg/dL)	142.5 (114.5–200)	142 (114–197)	0.97
TyG index	5.08 (4.9–5.26)	5.03 (4.9–5.25)	0.76
FG (mg/dL)	133.8 (67.2–222)	162 (142–196)	0.28
Hb1ac (%)	7.42 ± 0.2	7.58 ± 0.21	0.11
eGFR (mL/min/1.73 m^2^)	99.8 ± 10.3	96.9 ± 9.3	0.16
Creatinine (mg/dL)	0.84 (0.71–0.99)	0.82 (0.7–1.1)	0.64
ACR (mg/g)	5.76 (3.22–13.84)	9 (4.3–19)	0.08
Uric acid (mg/dL)	5.36 (4.35–6.89)	5.36 (4.45–6.11)	0.99
Calcium (mg/dL)	9.3 ± 0.34	9.36 ± 0.28	0.85
Phosphate (mg/dL)	3.55 ± 0.71	3.55 ± 0.42	0.98
NLR (/mL)	2.58 (1.62–3.89)	2.2 (1.8–3.52)	0.69
MLR (/mL)	0.33 (0.23–0.46)	0.32 (0.24–0.43)	0.86
hs-CRP (mg/L)	2.75 (1.25–5.9)	3.9 (2.1–6.8)	0.03
TNFα (pg/mL)	1.87 (1.34–2.72)	2.22 (1.57–3.1)	0.04
IL6 (pg/mL)	5.03 (2.48–8.97)	7.5 (3.51–12–25)	0.03
Klotho (pg/mL)	314.5 (6.15–562.81)	458.97 (275.2–667.2)	0.02
iFGF23 (pg/mL)	571.2 (444.5–675.4)	570.21 (421.3–707.3)	0.91
Medication			
Statin (%)	21 (70)	63 (61.2)	0.38
ACEI/ARB (%)	11 (36.7)	50 (48.5)	0.25

CAD, coronary artery disease; BMI, body mass index; SBP, systolic blood pressure; DBP, diastolic blood pressure; HDL-C high density lipoprotein cholesterol; LDL-C low-density lipoprotein cholesterol; TG, triglycerides; TyG, triglyceride glucose index; FG, fasting glucose; Hb1ac, glycated hemoglobin; eGFR, estimated glomerular filtrate rate; ACR, urine albumin-to-creatinine ratio; hs-CRP high sensitivity C-reactive protein, TNFα, tumor necrosis factor alpha; IL, interleukin; NLR, neutrophil lymphocyte ratio; MLR, monocyte lymphocyte ratio; iFGF23, intact fibroblast growth factor 23; ACEI, angiotensin converting enzyme inhibitor; ARB, angiotensin II receptor antagonist.

**Table 4 ijms-24-13456-t004:** Bivariate correlation study of serum Klotho protein levels with coronary artery stenosis determinations in patients with and without T2DM.

	Non-T2DM		T2DM	
	r	*p*	r	*p*
SSI	−0.3	0.675	0.233	0.007
Obs LCA (%)	−0.074	0.297	0.212	0.014
Obs RCA (%)	0.028	0.696	0.182	0.036
Obs LAD (%)	−0.063	0.379	0.152	0.082
Obs CA (%)	0.114	0.108	0.029	0.739

T2DM, type 2 diabetes mellitus; SSI, severity stenosis index; Obs, obstruction; LCA, left main coronary artery; RCA, right coronary artery; LAD, left anterior descending artery; CA, circumflex artery.

**Table 5 ijms-24-13456-t005:** Multiple stepwise regression analysis for stenosis severity index as the dependent variable in T2DM and non-T2DM subjects.

Stenosis Severity Index	Adjusted R^2^	ß	SE	t	*p*
Non-T2DM subjects	0.045				<0.05
TNFα (pg/mL)		0.130	0.228	1.837	0.048
ACR (mg/g)		0.129	0.178	1.849	0.046
T2DM subjects	0.153				<0.01
Dyslipidemia		0.206	3.852	2.469	0.015
Klotho (pg/mL)		0.184	0.007	2.201	0.030
ACR (mg/g)		0.228	0.225	2.735	0.007

T2DM, type 2 diabetes mellitus; TNFα, tumor necrosis factor alpha; ACR, urine albumin-to-creatinine ratio.

**Table 6 ijms-24-13456-t006:** Multivariate logistic regression analysis for the presence of significant CAD in the group of subjects with T2DM with serum Klotho levels as independent variable.

	Unadjusted		Model 1		Model 2		Model 3	
	OR (95% CI)	*p*	OR (95% CI)	*p*	OR (95% CI)	*p*	OR (95% CI)	*p*
Klotho (pg/mL)	1.003 (1.002–1.003)	<0.001	1.001 (1.0–1.003)	0.037	1.001 (1.0–1.003)	0.039	1.001 (1.0–1.003)	0.041

## Data Availability

Proposals relating to the data access should be directed to the corresponding authors.

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
