# Peer review of "Association of Klotho with Coronary Artery Disease in Subjects with Type 2 Diabetes Mellitus and Preserved Kidney Function: A Case-Control Study"

_ijms, 2023, doi:10.3390/ijms241713456_

Round 1
Reviewer 1 Report
The author conducted a cross-sectional case-control study that involved 133 individuals with type 2 diabetes (T2DM) and 200 non-diabetic patients of similar age, sex, and coronary artery disease (CAD) incidence who were undergoing non-emergency diagnostic coronary angiography. Using multiple regression analysis, the study found a positive relationship between serum Klotho levels and stenosis values, but only among subjects with T2DM (adjusted R2=0.153, P<0.01). Additionally, logistic regression analysis demonstrated a positive association between Klotho levels and the presence of significant CAD specifically within the group of T2DM patients (odds ratio: 1.001; P=0.041). These findings suggest that higher levels of circulating Klotho in individuals with T2DM and preserved kidney function are linked to the presence of significant CAD.
1. could you please provide information on how the severity of coronary artery disease is evaluated?
Author Response
Dear Reviewer 1.
Thanks for your kind words.
Answers to reviewer's comments:
- Could you please provide information on how the severity of coronary artery disease is evaluated?
Thanks for your comment. We have tried to clarify the methodology used to determine the severity of the stenosis.
"Coronary stenosis was determined by angiography using standard techniques. The assessment of stenosis was determined in four major epicardial arteries: left main coronary artery (LCA), left anterior descending artery (LAD), circumflex artery (CA), and right coronary artery (RCA). Significant CAD was defined as the presence of at least one lesion leading to ≥50% luminal diameter narrowing in any of the considered arteries. Additionally, a stenosis severity index (SSI) was defined as the average of the maximum stenosis values determined in each of those arteries."

Reviewer 2 Report
Dear Sirs,
the authors are dealing with a controversial issue such as Klotho in CAD. Their results seem interesting. However, I would propose to the authors to include in the discussion section and in the references more recent references, as this would make their manuscript more up to date and complete. In addition, what about the alterations in the gut microbiota associated with CAD? The authors could add a paragraph regarding a potential role of microbiota alterations in CAD. My recommendation would be accept after these minor additions.
Author Response
Dear Reviewer 2
Thanks for your kind words. We agree with you. The association of Klotho with vascular disease is very controversial.
- I would propose to the authors to include in the discussion section and in the references more recent references, as this would make their manuscript more up to date and complete.
We have included three new recent publications:
-
- Mao Q, Deng M, Zhao J, Zhou D, et al. Klotho ameliorates angiotension-II-induced endothelial senescence via restoration of autophagy by inhibiting Wnt3a/GSK-3β/mTOR signaling: A potential mechanism involved in prognostic performance of Klotho in coronary atherosclerotic disease. Mech Ageing Dev. 2023 Apr;211:111789.
- Kanbay M, Demiray A, Afsar B, Covic A, Tapoi L, Ureche C, Alberto Ortiz. Role of Klotho in the Development of Essential Hypertension. Hypertension. 2021;77:740–750
- Fard TK, Ahmadi R, Akbari T, Moradi N, Fadaei R, Fard MK, Fallah S. Klotho, FOXO1 and cytokines associations in patients with coronary artery disease. Cytokine. 2021 May;141:155443.
- In addition, what about the alterations in the gut microbiota associated with CAD? The authors could add a paragraph regarding a potential role of microbiota alterations in CAD.
Thanks for your interesting comment. We have added a short paragraph about this topic in the Discussion section and three references:
"An interesting question, particularly in subjects with T2DM, that has not yet been studied in depth is the potential relationship between the gut microbiota and Klotho. The gut microbiota has emerged as a significant contributory factor to age-related disease [42]. Moreover, the dysregulation in the composition and function of gut microbiota in patients with symptomatic atherosclerosis has been reported in many studies [43]. This dysreg-ulation has been proposed to trigger the immune system and cause systemic and local inflammation [44]. According to our hypothesis, this could cause a compensatory increase in Klotho production in patients with preserved kidney function"
